# Trigonally Distorted Hexacoordinate Co(II) Single-Ion Magnets

**DOI:** 10.3390/ma15031064

**Published:** 2022-01-29

**Authors:** Ivan Nemec, Ondřej F. Fellner, Berenika Indruchová, Radovan Herchel

**Affiliations:** 1Department of Inorganic Chemistry, Faculty of Science, Palacký University, 17. Listopadu 12, 77146 Olomouc, Czech Republic; ondrej.fellner01@upol.cz (O.F.F.); berenika.indruchova01@upol.cz (B.I.); radovan.herchel@upol.cz (R.H.); 2Central European Institute of Technology, Brno University of Technology, Purkynova 123, 61200 Brno, Czech Republic

**Keywords:** single-ion magnets, magnetic anisotropy, cobalt(II)

## Abstract

By simple reactions involving various cobalt(II) carboxylates (acetate and in situ prepared pivalate and 4-hydroxybenzoate salts) and neocuproine (neo), we were able to prepare three different carboxylate complexes with the general formula [Co(neo)(RCOO)_2_] (R = –CH_3_ for **1**, (CH_3_)_3_C– for **2**, and 4OH-C_4_H_6_– for **3**). The [Co(neo)(RCOO)_2_] molecules in the crystal structures of **1**–**3** adopt a rather distorted coordination environment, with the largest trigonal distortion observed for **1**, whereas **2** and **3** are similarly distorted from ideal octahedral geometry. The combined theoretical and experimental investigations of magnetic properties revealed that the spin Hamiltonian formalism was not a valid approach and the L-S Hamiltonian had to be used to reveal very large magnetic anisotropies for **1**–**3**. The measurements of AC susceptibility showed that all three compounds exhibited slow-relaxation of magnetization in a weak external static magnetic field, and thus can be classified as field-induced single-ion magnets. It is noteworthy that **1** also exhibits a weak AC signal in a zero-external magnetic field.

## 1. Introduction

Hexacoordinate Co(II) complexes often tend to possess very large magnetic anisotropies arising from the direct contribution of spin orbit coupling to the ground state. If the coordination geometry is regular octahedron, the ground state is ^4^T_1g_. Then, angular momentum is the main contributor to the zero-field splitting (ZFS), because the spin-orbit coupling operator transforms under t_1g_, thus it directly mixes with the ground state. Magnetic anisotropy is then very large and of the easy-plane type [1], the classic spin Hamiltonian description loses validity, and low-lying excited states appear [2]. The design of highly anisotropic magnetic molecules exploits changing of the regular octahedral geometry, which can be achieved either by elongation/compression of the metal–ligand bonds or by trigonal distortion. Both types of distortion can lead to double orbitally degenerate ground states (^4^E_g_ for compressed octahedron, D_4h_, ^4^E″ for trigonal prism, D_3h_), manifesting themselves again by a large contribution of spin-orbit coupling to the ground state. However, contrary to the ^4^T_1g_ state [3,4], the anisotropy of E-states is of the axial character [5]. Thus, Co(II) complexes with doubly degenerate ground states are highly interesting for synthesis of single-ion magnets (SIMs), which are a class of singe molecule magnets (SMMs) [6] containing only one paramagnetic center [7]. Among all SIMs based on 3d transition metal ions [8,9,10,11,12,13,14,15], those with trigonal prismatic environment around the Co(II) center have a special position, because they often exhibit slow relaxation of magnetization in the absence of an external magnetic field [16,17,18,19,20,21,22]. It must be noted that so-called zero-field SIMs (ZF-SIMs) are still very rare for complexes of 3d transition metals, which is because of the rather specific requirements needed for occurrence of ZF-SIMs; i.e., large and axial magnetic anisotropy with negligible rhombicity [23]. Thus, in line with the matter discussed above, the trigonal Co(II) complexes with doubly degenerate ground state are ideal candidates for ZF-SIMs. Besides this class of Co(II) compounds, there are only a few examples of 3d metal based ZF-SIMs: linear two-coordinate Fe(I) [24,25], Co(II) [10] complexes, pentacoordinate Fe(III) [26], or tetracoordinate Co(II) complexes [27,28,29,30,31,32].

In this paper, we focused our attention on trigonal distortion and, as the object of our research, we used very common carboxylate complexes [33] with one bidentate N-donor chelating ligand. Recently, we reported on two [Co(neo)(PhCOO)_2_] polymorphs (neo stands for neocuproine), which differed in their trigonal distortion and magnetic properties. Both compounds behaved as SIMs in a weak external magnetic field (*B* = 0.1 T), so-called field-induced SIMs [34]. In the present paper, we show that, for [Co(neo)(RCOO)_2_] complexes, we can achieve a significant change in the trigonality of the coordination polyhedron by variation of carboxylate ligands (RCOO–). We report on the synthesis, crystal structure, and thorough experimental and theoretical investigation of static and dynamic magnetic properties of three new carboxylate complexes with the general formula [Co(neo)(RCOO)_2_], where RCOO– represents carboxylate ligands (acetate (**1**), pivalate (**2**), and 4-hydroxybenzoate (**3**)).

## 2. Materials and Methods

### 2.1. Materials

Co(NO_3_)_2_·6H_2_O, Co(ac)_2_·4H_2_O, neocuproine, sodium pivalate hydrate, sodium 4-hydroxybenzoate, and solvents (MeOH, diethyl ether (Et_2_O)) were supplied by VWR International (Stříbrná Skalice, Czech Republic), Sigma-Aldrich (Prague, Czech Republic), Lach-Ner (Neratovice, Czech Republic), and Litolab (Chudobín, Czech Republic).

### 2.2. Synthesis

#### 2.2.1. Complex [Co(neo)(ac)_2_] (**1**)

To the solution of Co(ac)_2_·4H_2_O (0.48 mmol, 120 mg) in 5 mL of methanol, 100 mg of neocuproine (0.48 mmol) was added. The solution was ultrasonicated for 15 min. The violet solution was filtered through a paper filter and crystallized by slow diffusion of Et_2_O in a closed flask. Then, 130 mg of **1** was isolated by filtration (yield = 70%) as violet crystals, which were dried in a desiccator under reduced pressure (overnight). IR (ATR, v, cm^−1^): 408 w, 437 w, 550 w, 618 w, 675 m, 733 w, 778 w, 813 w, 846 w, 865 m, 937 w, 1007 w, 1037 w, 1162 w, 1226 w, 1299 w, 1383 m, 1421 s, 1440 s, 1501 m, 1556 s, 1593 m, 1617 w, 3001 w, 3055 w.

#### 2.2.2. Complexes [Co(neo)(piv)_2_] (**2**) and [Co(neo)(4OH-benz)_2_]·2CH_3_OH (**3**)

Compounds **2** and **3** were both prepared using the following method. To the solution of Co(NO_3_)_2_·6H_2_O (70 mg, 0.24 mmol) in 15 mL of methanol, 50 mg of neocuproine (0.24 mmol) was added. The solution was heated up to 50 °C, then 2 molar equiv. of the corresponding sodium salt was added (60 mg of sodium pivalate hydrate, **2**; 77 mg of sodium 4-hydroxybenzoate, **3**). The violet solution was filtered through a paper filter and crystallized isothermally. Then, 49 mg (yield = 43%) of **2** and 52 mg of **3** (yield = 36%) were obtained as violet crystals a couple of days later. The crystals were dried in a desiccator under reduced pressure (overnight).

**2**: IR (ATR, v, cm^−1^): 355 m, 407 w, 551 w, 609 w, 656 w, 681 w, 733 w, 776 w, 791 w, 810 w, 864 w, 900 w, 940 w, 1002 w, 1032 w, 1157 w, 1225 m, 1297 w, 1359 m, 1377 w, 1421 s, 1457 w, 1486 s, 1503 m, 1533 m, 1593 m, 2865 w, 2926 w, 2967 w.

**3**: IR (ATR, v, cm^−1^): 406 w, 504 w, 549 w, 630 m, 658 w, 701 w, 728 w, 783 m, 855 m, 1030 m, 1099 w, 1142 w, 1166 m, 1227 m, 1284 m, 1373 s, 1396 s, 1503 m, 1534 w, 1568 w, 1593 s, 2818 w, 2907 w, 2937 w, 3059 w, 3352 w, 3462 w.

Elemental analysis: **1**, Mr = 550.4, C_18_H_18_CoN_2_O_4_, found: C, 56.28 H, 4.74; N, 7.25, requires C, 56.11; H, 4.71; N, 7.27%, **2**, Mr = 567.9, C_24_H_30_CoN_2_O_4_, found: C, 61.38; H, 6.52; N, 5.90, requires C, 61.40; H, 6.44; N, 5.97%, **3** (composition calculated for [Co(neo)(4OH-benz)_2_]·1.8 CH_3_OH), *M*_r_ = 661.9, C_30_H_30_CoN_2_O_7_._8_, found: C, 59.46; H, 4.56; N, 4.97, requires C, 59.74; H, 4.91; N, 4.68.

### 2.3. General Methods

Elemental analysis was performed by a Flash 2000 CHNS Elemental Analyzer (Thermo Scientific, Waltham, MA, USA). A Jasco FT/IR-4700 spectrometer (Jasco, Easton, MD, USA) was used for the collection of the infrared (IR) spectra in the range of 400–4000 cm^−1^ using the attenuated total reflection (ATR) technique on a diamond plate. The static magnetic data were measured on powdered samples pressed into pellets using a PPMS Dynacool (Quantum Design Inc., San Diego, CA, USA). The dynamic magnetic data were measured on powdered samples pressed into pellets stabilized by eicosane using a MPMS XL-7 Quantum Design SQUID magnetometer (Quantum Design Inc., San Diego, CA, USA).

### 2.4. X-ray Crystallography

Data collection for **1**–**3** was done using the standard rotational method on a D8 Quest diffractometer equipped with a Photon 100 CMOS detector (Bruker, Billerica, MA, USA) using the Mo-Kα radiation (λ = 0.71073 Å). Data collection, data reduction, and cell parameters refinements were performed using the Bruker Apex III software package [35]. The structures were solved by SHELXT [36] and all non-hydrogen atoms were refined anisotropically on F^2^ using the full matrix least-squares procedure with Olex2.refine [37] in OLEX2 (version 1.5) [38]. All hydrogen atoms were found in differential Fourier maps and their parameters were refined using a riding model with *U*_iso_(H) = 1.2(CH) or 1.5(–CH_3_, –OH) *U*_eq_. The molecular structures and packing diagram were drawn with MERCURY [39]. 

Powder diffraction data (Appendix A) were collected using a MiniFlex600 (Rigaku) equipped with the Bragg–Brentano geometry, and with iron-filtered Cu Kα_1,2_ radiation.

## 3. Results and Discussion

### 3.1. Synthesis and Crystal Structure

Compound **1** was prepared by a direct reaction between Co(ac)_2_·4H_2_O and neo (molar ratio 1:1) in methanol. Compounds **2** and **3** were prepared in a very similar way, but Co(NO_3_)_2_·6H_2_O was used as the starting Co(II) compound and the corresponding sodium salts (NaRCOO) were used as a source of the carboxylate ligands. The reaction mixtures were prepared by dissolving Co(NO_3_)_2_·6H_2_O, neo, and NaRCOO in methanol (molar ratio of 1:1:2). These procedures proved to be good for the preparation of the crystalline products including single crystals useful for X-ray diffraction; however, the reaction yields were relatively low (30–50%).

Compounds **1**–**3** were isolated as violet crystals, which diffracted rather well, and we were able to determine crystal structures by single-crystal X-ray diffraction. The basic crystallographic data are summarized in Table 1. All compounds consist of the [Co(neo)(RCOO)_2_] complex molecules, only in **3** are two additional co-crystallized methanol molecules present in the asymmetric unit. The complex molecules are hexacoordinate, and all the ligands coordinate to the Co atom in a bidentate manner. The Co–N bonds adopt bond lengths between 2.08 and 2.13 Å, whereas the lengths of the Co–O bonds are more variable: 2.04–2.20 Å (Figure 1). The shapes of the coordination polyhedrons were evaluated using SHAPE algorithm and continuous shape measures (CSMs) [40]. We revealed that all the complex molecules possessed very large distortions from the ideal geometries. The coordination polyhedrons of **2** and **3** are closer to regular octahedral (O_h_) than trigonal prismatic geometry (D_3h_), with the following CSMs (O_h_, D_3h_): 7.650, 9.801, **2**; 6.352, 10.110, **3**. Complex **1** possesses the largest trigonal distortion and coordination geometry close to trigonal prism (11.893, 3.761, ESI Appendix A).

The non-covalent interactions in **1**–**3** are mainly of weak nature, including C–H···O, C–H···*π*, and *π*···*π* interactions of the neo aromatic rings. Of note here is the crystal structure of **3**, involving OH groups of the 4OH-benz ligands and co-crystallized molecules of methanol. These formed 2D networks of the molecules are interconnected by rather strong O–H···O hydrogen bonds with the donor···acceptor distances ranging between 2.65 and 2.85 Å (Appendix A). Even such relatively strong non-covalent contacts did not sufficiently stabilize the crystal structure of **3**. When the crystals were transferred outside of the mother-liquor, the solvent loss occurred upon drying and was accompanied by a loss of crystallinity and/or change in the unit cell parameters. Thus, at ambient conditions (*T* = 298 K, *p* = 1 atm), we were not able to unambiguously confirm the phase uniformity of **3**, because the X-ray diffraction pattern of the dried **3** differed slightly from the pattern calculated from the single-crystal structure. Nevertheless, the experiments performed using grinded crystals of **3** immersed in a highly viscose crystallographic oil showed that, after 1 h, the diffraction pattern underwent significant changes (Appendix A), which indicated that the batches of **3** we prepared were phase pure.

### 3.2. DC Magnetic Properties

The magnetic properties for **1**–**3** measured in a static magnetic field are depicted in Figure 2 as the temperature dependence of the effective magnetic moment and isothermal magnetization. Evidently, the profile of *μ*_eff_ versus *T* is varied within the series owing to the variation of the geometry of the coordination polyhedra, and hence the ligand field. As there are negligible intermolecular interactions, the decrease in *μ*_eff_ is attributed to the zero-field splitting/large magnetic anisotropy of these compounds. Indeed, this is also confirmed by the saturation values of *M*_mol_ of the isothermal magnetization curves well below the theoretical limit *g∙S*. 

Usually, the magnetic anisotropy is treated with the spin Hamiltonian comprising the zero-field splitting and Zeeman terms; however, hexacoordinate Co^II^ complexes with the shape of the chromophore close to octahedron or trigonal prism possess orbital angular momentum, hence the spin Hamiltonian is inappropriate. This was confirmed by ab initio calculations that predicted magnetic behavior based on E ground state for **1** and low lying (below 1000 cm^−1^) excited states for **2** and **3**, both contradicting the use of spin Hamiltonian formalism [41] (*vide infra*).

Therefore, the DC magnetic data were analysed with the L-S Hamiltonian based on Griffith and Figgis [42,43,44], which describes the splitting of the ^4^T_1g_ term originating from the ^4^F atomic term in lower symmetries than O_h_ as follows:(1)$$\hat{H}=-\alpha \cdot \lambda \left(\vec{S}\cdot \vec{L}\right)+\Delta_{\mathrm{ax}}\left({\hat{L}}_{z}^{2}-{\hat{L}}_{}^{2}/3\right)+\Delta_{\mathrm{rh}}\left({\hat{L}}_{x}^{2}-{\hat{L}}_{y}^{2}\right)+\mu_{B}\vec{B}\left(g_{e}\vec{S}-\alpha \vec{L}\right)$$

The splitting of the ^4^T_1g_ term is described by ∆_ax_ and ∆_rh_ parameters; *α* is an orbital reduction factor, *λ* is a spin-orbit coupling parameter, and *g*_e_ = 2.0023. Owing to the utilization of T_1_-P isomorphism, the angular orbital momentum *L* adopts the value of 1 with the effective Lande *g*-factor, *g_L_* = −1. The Hamiltonian acts on |*S*, *L*, *M_S_*, *M_L_* > functions with *M_L_* = 0, ±1 and *M_S_* = ±1/2, ±3/2 [45]. Next, the orbital reduction factor embodies two parameters, α = *Aκ*, where *A* is the Figgis coefficient of the configuration interaction resulting from the admixture of the excited terms reflecting the ligand field strength, and *κ* describes the lowering orbital contribution due to covalency of the metal–ligand bond. Moreover, the spin-orbit coupling parameter *λ* can be reduced in comparison with its free-ion value *λ*_0_ = −180 cm^−1^, which is attributable to the covalent character of the donor–acceptor bond.

The analysis encompasses both temperature- and field-dependent magnetic data and was done both for positive and negative values of ∆_ax_ with the help of a program POLYMAGNET [46]. However, only in the case of compound **2** were reasonably good fits achieved for both signs of ∆_ax_, whereas the negative sign of ∆_ax_ was found for **1** and **3**—Figure 2. The values of the fitted parameters are listed in Table 2. The negative values of the fitted Δ_ax_ parameters resulted in the easy axis type of magnetic anisotropy, as visualized in the three-dimensional plots of molar magnetization—Appendix A. Such a type of magnetic anisotropy is essential for the formation of the spin reversal barrier needed for the observation of the Orbach type mechanism of the slow relaxation of magnetization. Moreover, the respective energy levels in the zero magnetic field are also plotted for **1**–**3** in Appendix A.

### 3.3. AC Magnetic Properties

The SMMs are generally characterized by the AC susceptibility measurements, evidencing the slow relaxation of magnetization. Therefore, first, the AC data were measured in the zero static magnetic field for **1**–**3**, but only in the case of **1** did we observed a very weak signal of the imaginary susceptibility (*χ*″). Thus, the data were also measured for a varying static magnetic field, which resulted in a clear observation of non-zero *χ*″ susceptibility—Appendix A. Thus, the temperature and frequency AC susceptibility was measured at small *B*_DC_ = 0.1 T to suppress the quantum tunneling of magnetization for **1**–**3**, which revealed frequency-dependent maxima of the imaginary susceptibility, thus confirming the slow relaxation of magnetization—Figure 3, Figure 4 and Figure 5. The experimental data were analyzed with a program MIF&FIT [47] to the one component Debye’s model based on Equation (2):(2)$$\chi \left(\omega \right)=\frac{\chi_{T}-\chi_{S}}{1+{\left(i\omega \tau \right)}^{1-\alpha }}+\chi_{S}$$

Such an analysis resulted in the values of isothermal (*χ*_T_) and adiabatic (*χ*_S_) susceptibilities, relaxation times (*τ*), and distribution parameters (*α*) for **1**–**3** (Appendix A). For further analysis, only the data for which the fitted parameters were calculated with the standard deviation two times smaller than the value of the fitted parameter were considered. Afterwards, the temperature dependences of the relaxation times were analyzed with a model comprising the direct and Orbach mechanism:(3)$$\frac{1}{\tau }=AT+\frac{1}{\tau_{0}}\exp\left(U_{\mathrm{eff}}/kT\right)$$

The fitted data are displayed in Figure 3, Figure 4 and Figure 5 and the best-fit parameters are as follows: *A* = (6.52 ± 0.62) K^−1^s^−1^, *τ*_0_ = (1.361 ± 0.074) × 10^−7^ s, and *U*_eff_ = (37.7 ± 0.25) K for **1**; *A* = (471 ± 29) K^−1^s^−1^, *τ*_0_ = (6.2 ± 2.0) × 10^−6^ s, and *U*_eff_ = (19.0 ± 0.74) K for **2**; *A* = (2053 ± 49) K^−1^s^−1^, *τ*_0_ = (1.04 ± 0.31) × 10^−6^ s, and *U*_eff_ = (17.5 ± 1.1) K for **3.** The fitted values of *U*_eff_ = 26.2 cm^−1^ for **1**, 13.2 cm^−1^ for **2**, and 12.2 cm^−1^ for **3** are smaller than the energy gaps between the first and the second Kramers doublets of ∆ = 82 cm^−1^ for **1**, ∆ = 187/145 cm^−1^ for **2**, and ∆ = 181 cm^−1^ for **3** (Appendix A), but such a feature is typical for Co^II^ SMMs.

### 3.4. Theoretical Calculations

The electronic structure and magnetic properties of **1**–**3** were also studied by theoretical methods suitable for complexes with a multireference character. Therefore, the multireference calculations based on the state average complete active space self-consistent field (SA-CASSCF) [48] wave function method complemented by N-electron valence second-order perturbation theory (NEVPT2) [49,50] were conducted with an ORCA 5.0 computational package [51,52]. The experimental molecular structures were used, and just the positions of hydrogen atoms were normalized with Mercury software. The triple-ζ basis set def2-TZVP was used for all atoms except for carbon and hydrogen atoms, for which def2-SVP was applied [53]. The speed of the calculations was increased by using def2/J and def2-TZVP/C auxiliary basis sets [54,55], together with the chain-of-spheres (RIJCOSX) approximation to exact exchange [56,57] as implemented in ORCA. The active space was defined by seven electrons in five d-orbitals of Co^II^ (CAS(7e,5o)), and all possible multiplets, 10 quartets and 40 doublets, were involved in the calculations. Subsequently, the ab initio ligand field theory (AILFT) [58,59] was applied to calculate the splitting of d-orbitals, as shown in Figure 6. It is evident that splitting of d-orbitals for **1** is close to the pattern typical for a trigonal prism ligand field, whereas the splitting for **3** resembles a more typical octahedral ligand field. This nicely demonstrates the gradual change in the ligand field symmetry within the series of **1**–**3**. Next, the ^4^T_1g_ term is split within 0–2000 cm^−1^ for **2** and **3** owing to their deviations from ideal O_h_ symmetry, but, in the case of **1**, there is evidently a split ^4^E ground term (0–930 cm^−1^) belonging to D_3_ pseudosymmetry (Figure 6, middle). It can be also interpreted as a large distortion of O_h_ symmetry, evidenced by the large splitting of the ^4^T_1g_ term spanning energy interval of 0–5000 cm^−1^. This is also mirrored in the splitting of the six lowest Kramers doublets (Figure 6, right), which is largest for **1**.

The energies of the six lowest Kramers doublets were used for the analysis of the parameters of the Hamiltonian in Equation (1). Such a procedure we applied for the first time in the investigation of the above mentioned [Co(neo)(PhCOO)_2_] polymorphs [34] and then also for other Co^II^ complexes [60,61,62]. This procedure resulted in the values of *α*∙*λ*, ∆_ax_, ∆_rh_, which are listed in Table 2 and graphically presented in Appendix A. However, this procedure has one drawback, as it is not possible to determine the values of *α* and *λ* separately. To overcome this problem, we calculated temperature- and field-dependent magnetization data directly in an ORCA package resulting from CASSCF/NEVPT2 calculations. Subsequently, these magnetic data were fitted to Equation (1), but with fixed values of *α*∙*λ*, ∆_ax_, ∆_rh_, which enabled the determination of *α* and *λ*, as listed in Table 2. The respective fits are depicted in Appendix A. There are clear trends visible from the calculated values: *α* and ∆_ax_ are much larger for **1** than for **2**–**3**, evidencing the impact of the geometry change from the trigonal prism to the octahedral shape. We can also comment on the possible source of discrepancies between the values of the Hamiltonian parameters derived from fitting of the experimental data and from CASSCF/NEVPT2 calculations. Firstly, the fitting of magnetic data is limited only to the temperature interval of 1.9–300 K, whereas the energy levels from Equation (1) span an interval of up to several thousand cm^−1^/K, which means that the Boltzmann population of energetically higher Kramers doublets is negligible, and hence does not affect magnetic data. Secondly, the active space of theoretical calculations was limited to only five d-orbitals, thus the ligand-based orbitals are missing; however, such calculations are usually too demanding for such complexes with a larger number of atoms.

Furthermore, the SINGLE_ANISO module [63] now available in ORCA 5.0 was employed and the ab initio magnetization blocking barriers were computed for **1**–**3**, as displayed in Figure 7. The corresponding matrix element of the transversal magnetic moment between the ground states with opposite magnetization is close to the value of 0.5 for **2**–**3**, and thus is larger than 0.1, which suggests a large predisposition for the quantum tunneling of magnetization. On the contrary, the value of 0.08 for **1** is rather small. These results are in good agreement with the AC susceptibility data, where a weak non-zero out-of-phase signal in the zero magnetic field was observed only for **1**, and applying the static magnetic field was necessary to observe the slow relaxation of magnetization for **2** and **3**. The calculated energy barriers *U* are 140 cm^−1^ for **1**, 162 cm^−1^ for **2**, and 250 cm^−1^ for **3**, and reflect the alternation in the coordination polyhedron geometries.

## 4. Conclusions

In this report, we discussed the structure and magnetic properties of three compounds containing [Co(neo)(RCOO)_2_] molecules (neo = neocuproine, R = –CH_3_ for **1**, (CH_3_)_3_C– for **2**, and 4OH-C_4_H_6_– for **3**). All three complexes are hexacoordinate, with coordination environments rather distorted from ideal octahedron. Calculated continuous shape measures for **1** are close to trigonal prismatic geometry, whereas **2** and **3** adopt very trigonally distorted octahedral coordination environments. The magnetic properties were studied by DC and AC magnetometry, and it was revealed that very large magnetic anisotropy dominates the magnetic behavior of **1**–**3**. CASSCF/NEVPT2 calculations revealed that the origin of the large anisotropy in **1** is in the ^4^E ground state, whereas in **2** and **3**, the large magnetic anisotropy arises from low-lying excited ligand field terms owing to the distorted hexacoordinate coordination geometry. The largest intra-Kramers doublet splitting was calculated for **3**, whereas the smallest was calculated for **1**. This agrees rather well with the distortion from the regular octahedral geometry of the coordination polyhedron in these complexes (CSMs): 6.352 (in **3**) < 7.650 (in **2**) < 11.893 (in **1**). AC susceptibility measurements revealed that **1** exhibits the slow relaxation of magnetization even in the zero static magnetic field. However, the observed signal was rather weak, and this prevented us from performing further analysis. Thus, for all three compounds, AC susceptibility was measured in a static magnetic field (*B*_dc_ = 0.1 T) and the slow relaxation of magnetization was confirmed for all of them. Thus, **1**–**3** behave as field-induced single-ion magnets, where **1** has the largest *U*_eff_ = 38 K. SINGLE_ANISO calculations revealed that the probability of |3/2,+3/2> ↔ |3/2,−3/2> quantum tunneling is the lowest for **1**, whereas in **2** and **3**, it should be the dominant relaxation process in the absence of the external magnetic field. This agrees rather well with the experimental observations presented in this report and, furthermore, it underlines the importance of trigonal coordination geometry (as in **1**) for the preparation of Co(II) zero-field SIMs.

## Figures and Tables

**Figure 1 materials-15-01064-f001:**
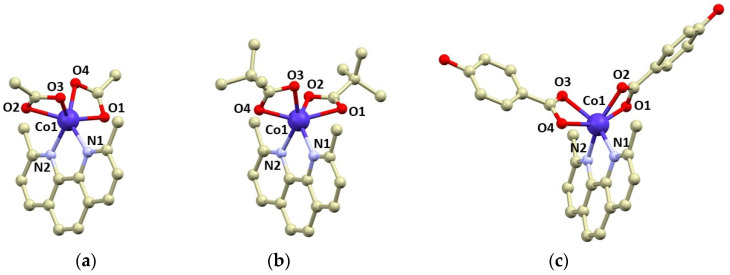
Molecular structures of complex molecules in the crystal structure of **1** (**a**), **2** (**b**), and **3** (**c**). Selected bond lengths (in Å): Co1–N1 = 2.105(2) in **1**, 2.133(2) in **2**, 2.130(4) in **3**; Co1–N2 = 2.125(2) in **1**, 2.118(2) in **2**, 2.078(3) in **3**; Co–O1 = 2.175(2) in **1**, 2.198(9) in **2**, 2.102(3) in **3**; Co–O2 = 2.163(2) in **1**, 2.150(9) in **2**, 2.160(4) in **3**; Co–O3 = 2.160(2) in **1**, 2.149(3) in **2**, 2.196(4) in **3**; Co–O4 = 2.104(2) in **1,** 2.119(6) in **2**, 2.131(3) in **3**.

**Figure 2 materials-15-01064-f002:**
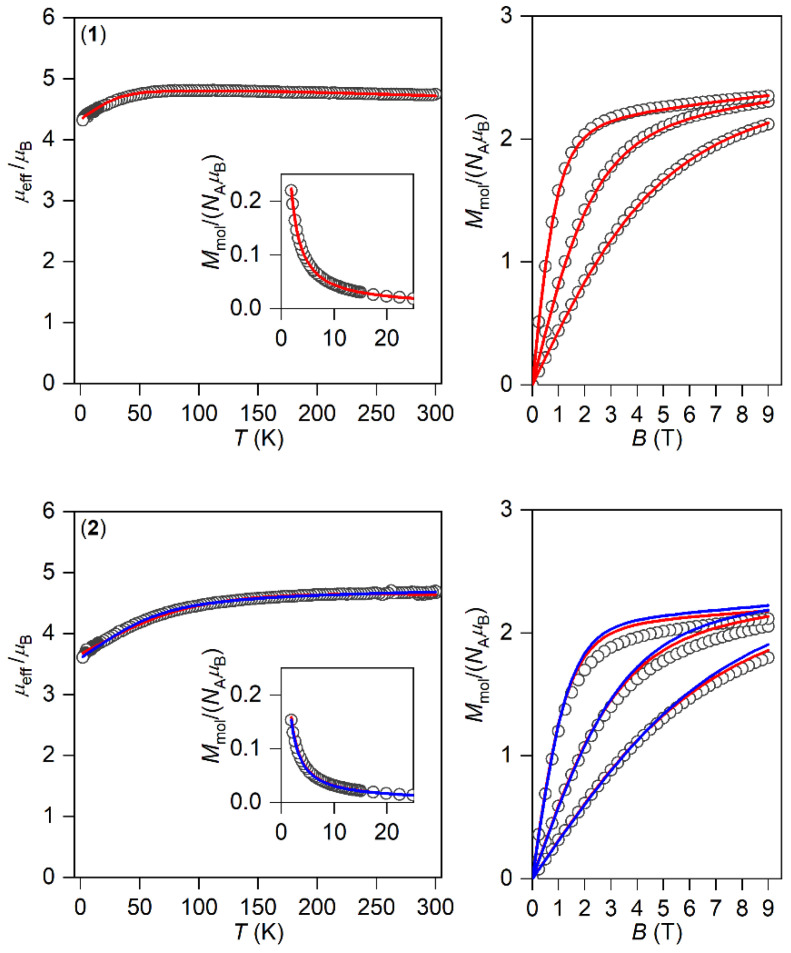
The DC magnetic data for **1**–**3** shown as the temperature dependence of the effective magnetic moment and isothermal molar magnetization measured at *T* = 2, 5, and 10 K. The empty symbols represent the experimental data; the full lines represent the fitted data using Equation (1) with the Hamiltonian parameters in Table 2, and the red and blue lines correspond to the negative and positive value of ∆_ax_, respectively.

**Figure 3 materials-15-01064-f003:**
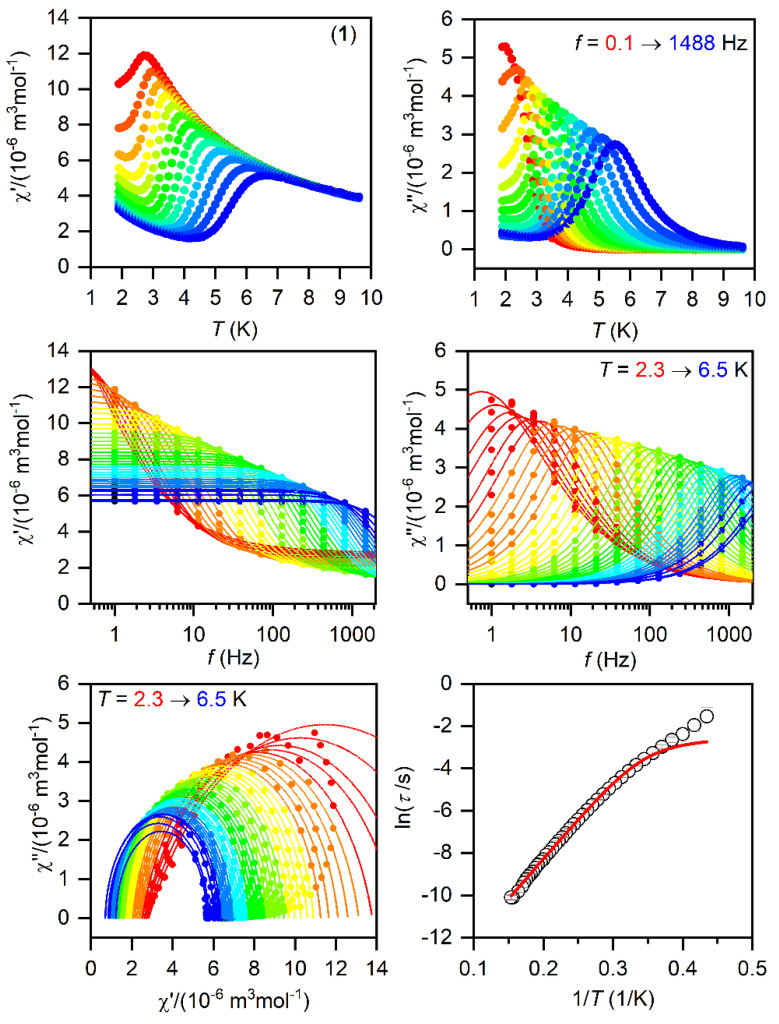
The AC magnetic data for **1**. Temperature dependence of the real (χ′) and imaginary (χ″) components of the AC susceptibility at the applied external magnetic field *B*_DC_ = 0.1 T for frequencies from 1 to 1500 Hz (full lines are only guides for eyes) (**top**). Frequency dependence of χ′ and χ″ molar susceptibilities fitted with one-component Debye’s model using Equation (2) (full lines) (**middle**). The Argand (Cole-Cole) plot with full lines fitted with Equation (2) and, on the right, the fit of resulting relaxation times *τ* with the direct + Orbach relaxation processes (red line) using Equation (3) (**bottom**).

**Figure 4 materials-15-01064-f004:**
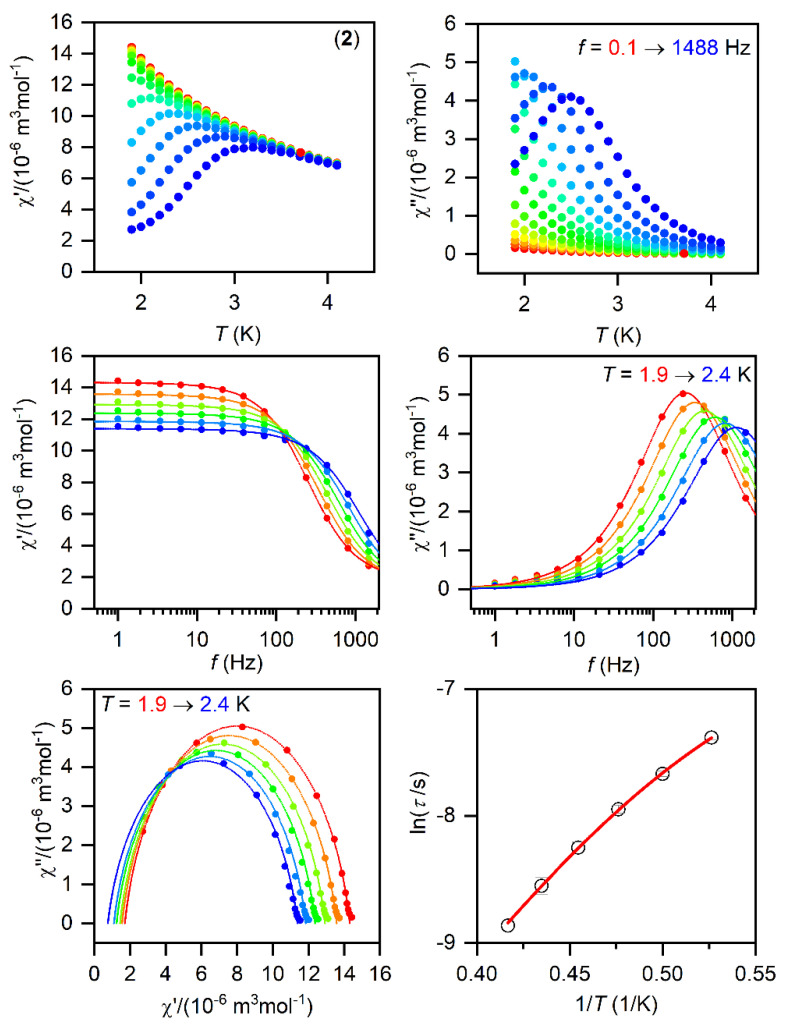
The AC magnetic data for **2**. Temperature dependence of the real (χ′) and imaginary (χ″) components of the AC susceptibility at the applied external magnetic field *B*_DC_ = 0.1 T for frequencies from 1 to 1500 Hz (full lines are only guides for eyes) (**top**). Frequency dependence of χ′ and χ″ molar susceptibilities fitted with one-component Debye’s model using Equation (2) (full lines) (**middle**). The Argand (Cole-Cole) plot with full lines fitted with Equation (2) and, on the right, the fit of resulting relaxation times *τ* with the direct + Orbach relaxation processes (red line) using Equation (3) (**bottom**).

**Figure 5 materials-15-01064-f005:**
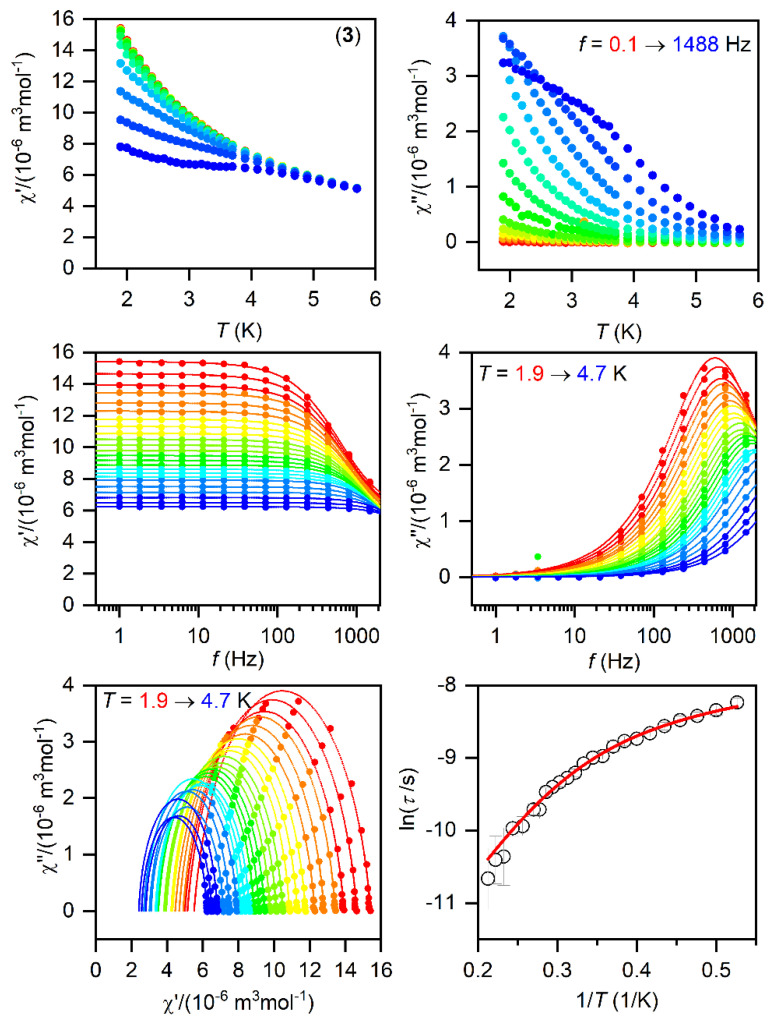
The AC magnetic data for **3**. Temperature dependence of the real (χ′) and imaginary (χ″) components of the AC susceptibility at the applied external magnetic field *B*_DC_ = 0.1 T for frequencies from 1 to 1500 Hz (full lines are only guides for eyes) (**top**). Frequency dependence of χ′ and χ″ molar susceptibilities fitted with one-component Debye’s model using Equation (2) (full lines) (**middle**). The Argand (Cole-Cole) plot with full lines fitted with Equation (2) and, on the right, the fit of resulting relaxation times *τ* with the direct + Orbach relaxation processes (red line) using Equation (3) (**bottom**).

**Figure 6 materials-15-01064-f006:**
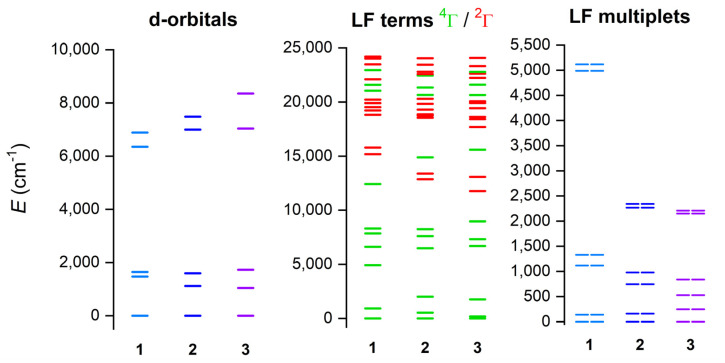
The outcome of the CASSCF/NEVPT2 calculations for complexes of **1**–**3**. Plot of the d-orbitals splitting calculated by ab initio ligand field theory (AILFT) (**left**), low-lying ligand-field terms (LFT) (**middle**), and ligand-field multiplets (LFM) (**right**). Note: different multiplicities of LFT are shown in a different color.

**Figure 7 materials-15-01064-f007:**
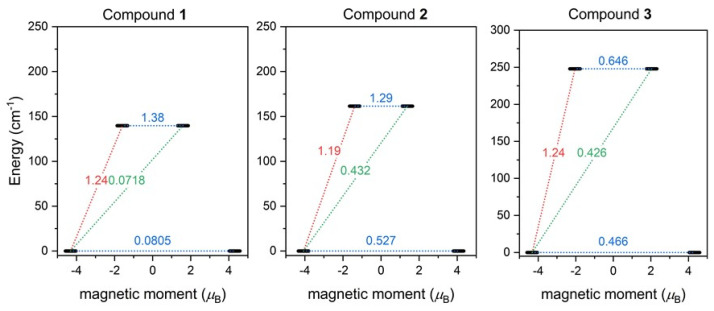
The outcome of SINGLE_ANISO CASSCF/NEVPT2 calculations for **1**–**3**. The numbers presented in the plots represent the corresponding matrix element of the transversal magnetic moment (for values larger than 0.1, an efficient relaxation mechanism is expected). Dashed lines refer to (temperature-assisted) quantum tunneling (blue), Orbach/Raman mechanisms (red), and direct/Raman mechanisms (green) [64].

**Table 1 materials-15-01064-t001:** Crystallographic data for **1**–**3**.

	1	2	3
Formula	C_18_H_18_CoN_2_O_4_	C_24_H_30_CoN_2_O_4_	C_30_H_30_CoN_2_O_8_
Formula weight	385.27	469.43	605.49
Crystal system	monoclinic	triclinic	monoclinic
Space group	*C*2/*c*	$P\bar{1}$	*P*2_1_/*c*
Cell parameters			
*a*/Å	14.0976(17)	9.5282(13)	9.922(4)
*b*/Å	9.4555(12)	9.8422(13)	19.083(7)
*c*/Å	25.255(4)	14.4559(16)	15.191(6)
*α*/deg	90	87.490(4)	90
*β*/deg	95.643(12)	80.364(4)	91.472(14)
*γ*/deg	90	62.028(4)	90
*V*/Å^3^	3350.1(8)	1179.5(3)	2875(2)
*Z*	8	2	4
Density, Dc/g cm^−^^3^	1.528	1.322	1.399
Abs. coefficient/mm^−^^1^	1.050	0.759	0.650
Data/restraints/param	2947/0/230	4629/566/422	5057/0/378
R1 ^a^, wR_2_ ^b^ (all data)	0.0412, 0.0730	0.0594/0.1068	0.1200/0.1847
R_1_ ^a^, wR_2_ ^b^ [I > 2 s(I)]	0.0296, 0.0700	0.0411/0.0999	0.0611/0.1614
Goodnes of fit	1.073	1.034	1.064
CSD number	2,126,276	2,126,278	2,126,275

^a^ R_1_ = ∑ (|F_o_| – |F_c_|)/∑|F_o_|, ^b^ wR^2^ = {∑[w(F^2^_o_ – F^2^_c_)^2^]/∑[w(F^2^_o_)^2^]}^1/2^.

**Table 2 materials-15-01064-t002:** The parameters of the Hamiltonian in Equation (1) derived from the experimental and calculated data.

Parameters	1	2 *^a^*	3
the analysis of DC data			
∆_ax_ (cm^−1^)	−3317	−523/810	−1051
∆_rh_ (cm^−1^)	−133	−23.6/32.5	−39.6
α	1.66	1.04/1.25	1.21
*λ* (cm^−1^)	−75.8	−167/−180	−151
the analysis of CASSCF/NEVPT2 energy levels			
∆_ax_ (cm^−1^)	−4322	−1703	−1641
∆_rh_ (cm^−1^)	−465	−276	−65.0
α∙*λ* (cm^−1^)	−303	−251	−256
the analysis of CASSCF/NEVPT2 magnetic data			
α	1.98	1.55	1.67
*λ* (cm^−1^)	−153	−162	−153

*^a^* The parameters corresponds to the best-fit for the negative and positive value of ∆_ax_.

## Data Availability

All data contained within the article.

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
