# Peer review of "Trigonally Distorted Hexacoordinate Co(II) Single-Ion Magnets"

_materials, 2022, doi:10.3390/ma15031064_

Round 1

Reviewer 1 Report

In this paper, the author presented the synthesis, crystal structure and magnetic properties of three mononuclear Co(II) complexes containing neocuproine and carboxylates. All compounds show field-induced slow magnetic relaxation behavior with different energy barriers and the magneto-structural relationship is also discussed. From my points of view, neither their crystal structures nor the dynamic magnetic properties are interesting. However, the work is completely done and the paper is properly organized. Thus, I think the paper is suitable for the publication in Materials. Some minor points are listed as below.

(1) The Elemental Analysis results for all the complexes are required.

(2) The error for the cell parameter are too large.

(3) Why did the author only fit the ln(tau) vs 1/T plots with direct and Orbach process? Have they attempted to fit the data just with other process, such as Raman and Orbach?

(4) By looking at the imaginary component of the susceptibility versus frequency plots, one clearly sees that the some points for compounds 1-3 do not present a maximum. As such, the authors have not measured the relaxation times at these temperatures  and including them in the relaxation profile fitting is wrong.

(5) The author should cite some important reference for the Co(II) Single-ion Magnets (Inorg. Chem. 2020, 59, 8505−8513; Chem. Sci. 2013, 4, 1802−1806).

Author Response

Thank you for your report!

Reviewer 3 Report

The authors report three mononuclear complexes based on cobalt(II) carboxylates (acetate, pivalate, and 4-hydroxybenzoate) and neocuproine. All the complexes are thoroughly characterized from the structural point of view. Trigonally distorted geometry is observed around the hexacoordinated Co(II) ions for all the complexes. The presence of slow relaxation of magnetization is confirmed from the magnetic investigation. Detail CASSCF/NEVPT2 calculations reveal the origin of the large magnetic anisotropy. All the complexes can be treated as excellent examples of field-induced single-ion magnets. I believe the overall work has significant importance. So, publication in Materials as an article can be recommended, but the following minor issues should be addressed before acceptance.

  1. Authors could add few recent literatures as references for low coordinate cobalt based SIMs in the introduction part.
  2. In line 63, neo should be replaced by neocuproine.
  3. Please include Figure S1 with better resolution in the revised manuscript.
  4. Please complete the bracket with lambda in all figure captions for pxrd.
  5. There are few sentences with minor grammatical errors and typos. Please have a look on those.

Author Response

Thank you for your review!
